# Advancing Public Health Surveillance: Integrating Modeling and GIS in the Wastewater-Based Epidemiology of Viruses, a Narrative Review

**DOI:** 10.3390/pathogens13080685

**Published:** 2024-08-14

**Authors:** Diego F. Cuadros, Xi Chen, Jingjing Li, Ryosuke Omori, Godfrey Musuka

**Affiliations:** 1Digital Epidemiology Laboratory, Digital Futures, University of Cincinnati, Cincinnati, OH 41221, USA; chen2x6@ucmail.uc.edu; 2Department of Geography and GIS, University of Cincinnati, Cincinnati, OH 41221, USA; 3Department of Land Resources Management, China University of Geosciences, Wuhan 430074, China; lijingjing@cug.edu.cn; 4Division of Bioinformatics, International Institute for Zoonosis Control, Hokkaido University, Sapporo 002-8501, Japan; omori@czc.hokudai.ac.jp; 5International Initiative for Impact Evaluation, Harare 0002, Zimbabwe; gmusuka@3ieimpact.org

**Keywords:** wastewater-based epidemiology, geographic information systems, mathematical modeling, remote sensing, public health surveillance, COVID-19

## Abstract

This review article will present a comprehensive examination of the use of modeling, spatial analysis, and geographic information systems (GIS) in the surveillance of viruses in wastewater. With the advent of global health challenges like the COVID-19 pandemic, wastewater surveillance has emerged as a crucial tool for the early detection and management of viral outbreaks. This review will explore the application of various modeling techniques that enable the prediction and understanding of virus concentrations and spread patterns in wastewater systems. It highlights the role of spatial analysis in mapping the geographic distribution of viral loads, providing insights into the dynamics of virus transmission within communities. The integration of GIS in wastewater surveillance will be explored, emphasizing the utility of such systems in visualizing data, enhancing sampling site selection, and ensuring equitable monitoring across diverse populations. The review will also discuss the innovative combination of GIS with remote sensing data and predictive modeling, offering a multi-faceted approach to understand virus spread. Challenges such as data quality, privacy concerns, and the necessity for interdisciplinary collaboration will be addressed. This review concludes by underscoring the transformative potential of these analytical tools in public health, advocating for continued research and innovation to strengthen preparedness and response strategies for future viral threats. This article aims to provide a foundational understanding for researchers and public health officials, fostering advancements in the field of wastewater-based epidemiology.

## 1. Introduction

### 1.1. Basic Definition

Wastewater-based epidemiology (WBE) represents a transformative methodological approach in public health, employing the analysis of wastewater to detect and monitor the presence of infectious agents, including viruses, bacteria, and other pathogens, within a defined population. This innovative approach relies in the natural collection of biological and chemical excretions from humans into wastewater systems, providing a unique and comprehensive lens through which to observe the health dynamics of entire communities [1,2,3]. Unlike traditional disease surveillance that requires individual testing and reporting, WBE acts as a collective diagnostic tool. It captures health-related indicators from a broad population base, continuously aggregating data from the daily flow of wastewater. This method thus eliminates the need for individual participation, bypassing potential barriers such as accessibility, stigma, and logistical constraints associated with personal health testing [2,4].

### 1.2. Historical Perspective—Origins and Early Uses

The origins of WBE trace back to the detection of environmental pollutants. Initially, scientists used wastewater analysis to identify industrial chemicals, heavy metals, and other contaminants, focusing on environmental health and safety. Wastewater was recognized as a reflection of community health, revealing consumption patterns and exposure risks through biomarkers like pharmaceuticals and personal care products [1,2,5,6,7,8,9,10].

WBE’s role in poliovirus eradication is particularly noteworthy. It identified hidden reservoirs of the virus, aiding global eradication efforts. The method detected both wild and vaccine-derived polioviruses, guiding targeted immunization campaigns and monitoring for potential outbreaks [11,12]. The scope of WBE expanded to include other pathogens like norovirus, hepatitis A and E, and bacterial pathogens such as *Salmonella* and *Escherichia coli*. PCR and NGS played key roles in this expansion, enabling the sensitive detection of pathogens and antimicrobial resistance genes, which is crucial for managing public health threats [13,14,15].

WBE has proven to be an invaluable early warning system, capable of detecting disease outbreaks before clinical cases emerge. By continuously monitoring wastewater for genetic markers or other indicators of pathogens, WBE allows public health officials to identify upticks in infection rates in real-time, often days or even weeks before individuals begin to exhibit symptoms [16,17]. This early detection capability is crucial for enabling proactive public health interventions. When WBE identifies an emerging pathogen in a community’s wastewater, health officials can immediately implement targeted measures such as increased diagnostic testing, quarantine protocols, vaccination drives, or public awareness campaigns. This precise targeting of resources ensures that interventions are efficient and effective, concentrating efforts where they are most needed to prevent the wider spread of the disease [2,7].

Furthermore, the ability of WBE to provide ongoing surveillance supports the continuous assessment of public health strategies. As interventions are applied, WBE can track changes in the viral or bacterial load in the community, offering feedback on the effectiveness of these measures and guiding adjustments as necessary. This iterative process ensures that public health responses remain agile and responsive to evolving threats, ultimately minimizing the health, social, and economic impacts of infectious diseases [3,4].

### 1.3. COVID-19 Pandemic—A Milestone

The COVID-19 pandemic marked a watershed moment for WBE, highlighting its critical role in providing early warnings and ongoing monitoring of the spread of SARS-CoV-2 among populations. As the virus rapidly spread globally, traditional surveillance systems were often overwhelmed, leading to significant gaps in tracking and responding to the outbreak. WBE emerged as a pivotal tool, offering a non-invasive, comprehensive, and timely method of detecting the presence of the virus in communities, even before clinical cases surged [17,18].

During the pandemic, WBE was instrumental in tracking the spread of SARS-CoV-2, often revealing the presence of the virus in wastewater samples days or weeks before a rise in clinical cases. This early detection capability enabled public health officials to preemptively tighten or adjust containment measures, such as lockdowns and social distancing, in specific areas based on the viral load detected in the wastewater. This approach allowed for a more dynamic and targeted response, helping to curb the spread of the virus and to manage health care resources more effectively [19,20,21]. 

While WBE has shown great promise in monitoring SARS-CoV-2, it is important to acknowledge that standardized protocols for sampling, virus concentration, and detection methods are not yet universally established. Current practices vary significantly between laboratories, which can affect the comparability and reliability of the results. This underscores the need for ongoing efforts to develop and harmonize methodologies across different settings. Establishing standardized protocols is crucial for improving the consistency and accuracy of WBE data, ultimately enhancing its utility in public health surveillance and response efforts. Researchers and public health practitioners collaborated globally to refine techniques such as qPCR and NGS to accurately detect and quantify the virus in wastewater. These standardized methods enhanced the reliability of WBE, ensuring consistent results across different laboratories and regions [22,23].

Other recent studies have underscored the importance of environmental health management in controlling the COVID-19 pandemic, emphasizing the need for robust surveillance systems and effective public health interventions. Key findings highlight the persistence of SARS-CoV-2 in various environmental media, such as air, surfaces, water, and wastewater, necessitating comprehensive strategies for infection control [24]. WBE has proven valuable for monitoring COVID-19 prevalence, providing early warnings of outbreaks and facilitating targeted public health responses. 

Polo et al. [25] highlight the crucial role of WBE in monitoring SARS-CoV-2, bridging the gap between traditional clinical surveillance and community-wide monitoring. They discuss the challenges of classical surveillance methods, such as limited diagnostic capacity and difficulty in capturing asymptomatic cases and emphasize the capability of WBE for early outbreak detection and better assessment of community infection levels. Their study outlines steps for implementing effective WBE, including sample collection, viral concentration, and ethical considerations, underscoring its potential to provide timely public health data and support pandemic response strategies. Similarly, Ciannella et al. [26] conducted a systematic review evaluating WBE for COVID-19 surveillance, identifying a lack of standardization in analytical methods and emphasizing the importance of correlating clinical data with wastewater findings. Despite these challenges, the authors highlight the effectiveness of WBE in monitoring SARS-CoV-2 circulation and its utility in supplementing clinical testing for comprehensive public health surveillance. Likewise, a model presented in another study estimates COVID-19 infections from SARS-CoV-2 RNA concentrations in wastewater using samples collected from sewersheds in South Carolina [27]. The model revealed that the number of unreported COVID-19 cases was approximately 11 times higher than that of confirmed cases, demonstrating the utility of wastewater surveillance in providing real-time, cost-effective monitoring of community-level transmission. Similarly, a review conducted by Gonçalves et al. [28] evaluated centralized and decentralized WBE approaches, highlighting their combined advantages for comprehensive surveillance. The review suggests that decentralized WBE can identify outbreak hotspots in specific locations like hospitals and schools, enabling the use of targeted public health interventions to control virus spread. These studies collectively underscore the transformative potential of WBE in managing the COVID-19 pandemic by providing early warnings, guiding public health interventions, and improving decision-making processes.

Furthermore, the integration of WBE data with other epidemiological and mobility data played a crucial role in guiding public health interventions. The data from wastewater surveillance were used to map hotspots of viral transmission, assess the effectiveness of public health measures, and make informed decisions about reopening schools, businesses, and public spaces. This holistic approach demonstrated the potential of WBE, not just as a surveillance tool, but as an integral part of public health strategy in managing pandemics and other infectious diseases [1,29,30].

## 2. Data Collection and Significance in Public Health

### 2.1. Data Collection and Integration with Other Data Sources

Wastewater-based epidemiology (WBE) involves systematic sampling across sewage systems, from residential, commercial, and industrial zones, ensuring representativeness and capturing diverse pathogenic profiles in the community. Techniques like PCR and NGS enable precise pathogen detection and quantification by identifying viral and bacterial genetic markers, providing a snapshot of community health [7,20,31]. The extensive surveillance scope of WBE, including small neighborhoods to entire regions, is supported by a comprehensive sewage network, allowing for the detection of both localized and widespread epidemiological trends [32,33]. Its versatility spans various pathogens and chemical contaminants, making it invaluable in tracking diseases like COVID-19, norovirus, hepatitis, and environmental and pharmaceutical exposures [2,34].

The non-invasive nature of WBE facilitates anonymous, continuous population health monitoring without requiring individual consent, thus overcoming traditional surveillance limitations related to privacy and participation barriers. By utilizing existing wastewater infrastructure, WBE efficiently assesses health trends, enabling swift public health responses and strategic resource allocation. This method surpasses traditional approaches in regards to cost-effectiveness and operational efficiency [2,32,35,36,37].

When integrated with clinical surveillance, demographic data, and environmental monitoring, WBE offers a comprehensive perspective on public health. This integration enriches our understanding of disease dynamics, linking increases in wastewater pathogen loads with clinical case upsurges for timely disease impact assessments [38,39]. Demographic insights allow for targeted interventions in vulnerable populations, ensuring equitable resource distribution [29,40]. Environmental monitoring connects pathogen concentration changes in WBE to environmental conditions, aiding in outbreak prediction based on factors like weather and water quality [41,42]. Advanced analytics and modeling, including machine learning, enhance WBE data interpretation, enabling public health officials to forecast outbreaks, adjust strategies, and evaluate the effectiveness of interventions [32,43]. This integrated approach represents a shift in public health surveillance, providing a holistic, responsive framework for managing health trends in populations.

### 2.2. Real-Time Surveillance and Cost Effectiveness

WBE stands as a real-time, cost-effective surveillance method that significantly enhances the detection and management of public health threats. Its continuous wastewater collection allows near real-time pathogen monitoring, serving as an early warning system that can detect infectious signals days to weeks before clinical cases arise. This early detection is crucial for initiating swift containment measures, optimizing medical resource allocation, and implementing timely interventions to prevent disease spread [17,18,44].

The rapid insights produced by WBE are invaluable during fast-moving outbreaks, enabling proactive responses such as enhanced surveillance, increased testing, and quarantine measures to prevent larger crises. This approach proved effective during the COVID-19 pandemic, guiding lockdown and reopening strategies based on community wastewater viral loads [19,20]. Additionally, WBE is cost-effective compared to traditional epidemiological methods reliant on individual testing. It reduces labor, materials, and infrastructure costs by using existing wastewater treatment facilities, integrating routine sampling into municipal water treatment processes. This maximizes public health surveillance while minimizing financial burdens, enhancing the agility and effectiveness of public health responses [2,32,45].

### 2.3. Equitable Public Health Monitoring and Support of Public Health Strategies and Policy Development

WBE serves as a crucial tool for equitable public health monitoring by capturing data from all socioeconomic segments within wastewater catchment areas. This approach includes underserved and vulnerable populations often missed by traditional public health surveillance due to barriers like limited healthcare access and privacy concerns. By analyzing community-wide wastewater samples, WBE ensures the comprehensive reflection of health status, including that of those individuals not visible in clinical data [1,3,45].

The inclusive data collection used in WBE is vital for identifying health disparities and tailoring interventions to community needs. Detecting pathogen level variations across different areas helps pinpoint neighborhoods or groups with higher infection rates or exposure to harmful substances. This insight enables public health officials to deploy targeted interventions like vaccination drives and enhanced sanitation, reducing inequities and improving community health outcomes [20,35]. Furthermore, WBE data inform public health strategies and policy development, providing a real-time snapshot of community pathogen loads to assess and adjust public health measures [18,23]. This feature is critical in pandemic preparedness, enhancing the ability to respond swiftly to new threats by integrating WBE with clinical and epidemiological data, creating a robust framework for managing public health crises [17,32].

The global reach of WBE, from high-income to low-resource settings, underscores its adaptability and importance in global health surveillance. Its non-invasive, scalable approach allows for rapid resource deployment in response to emerging health threats, enhancing global health security. Integrating WBE data with other health data sources enhances public health decision making, providing insights into disease spread, contributing to international disease control, and aiding in future health challenge anticipation [18,30,32].

In summary, WBE stands as a vital tool in global health security, especially in settings where traditional surveillance systems may be inadequate or absent. Its ongoing monitoring capabilities aid in long-term health planning, including infrastructure development, environmental health initiatives, and preventive health measures, ensuring that WBE remains at the forefront of efforts to track the emergence and re-emergence of pathogens and to prepare appropriate responses to safeguard public health.

## 3. Time-Dependent Modeling Techniques in Wastewater Surveillance

### 3.1. Basic Principles of Modeling in Wastewater Surveillance

Modeling in WBE is essential for translating pathogen concentration data into actionable public health insights. Through statistical and computational models, public health officials can estimate community infection rates, predict trends, and assess the impact of interventions, enhancing decisions to protect community health. These models consider factors like wastewater dilution, viral RNA decay, and population metrics to accurately estimate virus prevalence, providing a comprehensive view of infection dynamics, including asymptomatic and undiagnosed cases [17,18,20,23].

Additionally, modeling predicts pathogen trends, facilitating early intervention planning and timely implementation of measures like lockdowns and vaccination campaigns before clinical cases rise. This predictive ability is crucial for preemptive public health responses. Models also evaluate intervention impacts by comparing changes in wastewater pathogen levels with expected trends, helping to determine the effectiveness of strategies like social distancing and vaccination in disease spread reduction [2,19,30,46]. Overall, WBE modeling provides a robust framework for understanding and managing public health threats, guiding informed strategies and interventions.

Wastewater systems are usually centralized networks. The sewage network is combined with storm water drainage systems in many regions, which may lead to combined sewer overflow issues [47]. The dilution and transmission of viral loads in wastewater with independent or combined sewer systems can be modeled using process-based hydrologic–hydraulic modeling frameworks. One widely used model for water quantity and quality simulation in urban environments is the storm water management model (SWMM). The model is capable of simulating the transport of viral loads in sewer systems based on a one-dimensional advection–dispersion equation that has been successfully applied in WBE [47,48].

### 3.2. Mathematical Models in Wastewater Surveillance

The detection of pathogens from wastewater can qualitatively reveal the existence of epidemic outbreaks in the focal population (qualitative approach). The quantitative measurement of the concentration of pathogens can take the understanding of epidemic one step further. Comparing the pathogenic load in wastewater between different sampling time points or locations can provide the time-series or spatial trend of epidemics. Interpreting the comparison of pathogen loads in wastewater samples requires a mathematical model describing the quantitative relationship between the epidemic and the pathogen load in wastewater [49,50]. For instance, a statistic characterizing an epidemic could reflect the number of infected individuals, and the relationship between the pathogen load in wastewater and the number of infected individuals is derived from the mathematical model describing how the pathogen is shed from the infected individuals and concentrated/decayed in water. If this relationship is quantitatively accurate, the absolute number of infected individuals can be estimated from the pathogen load in wastewater (quantitative approach) [51,52]. Even if the relationship can only estimate the relative statistics of an epidemic, e.g., the relative number of infected individuals (semi-quantitative approach), this relationship is quite informative regarding the trend of the epidemic. The difficulty in deriving a quantitative relationship between the epidemic dynamics and the pathogen load in wastewater has been noted, and the use of artificial intelligence is expected to solve this challenge [53,54]. A short summary of several studies implementing different modeling methods used in wastewater surveillance is included in Table 1.

### 3.3. Challenges and Limitations 

Modeling approaches in WBE provide insights into infectious disease spread within communities, but accurately modeling virus concentrations is marked by challenges impacting the reliability of WBE. Variability in viral shedding rates due to infection stage and immune responses introduces uncertainty in regards to estimating community-wide infection rates [58,59]. Likewise, environmental degradation of viral RNA, influenced by temperature, pH, and other contaminants, complicates the quantification of virus levels. Wastewater treatment processes like dilution and chemical treatments further alter viral concentrations, making direct correlations with community infection rates challenging [60,61].

Model validation and calibration with actual surveillance data are essential to ensure reliability. Rigorous testing and adjustments based on wastewater and clinical data refine model parameters and enhance predictive accuracy. Uncertainties in parameters like viral decay rates require sensitivity analysis to identify influential factors and areas requiring robust data collection for improved model reliability [62].

## 4. Spatial Analysis in Wastewater Surveillance

Spatial analysis in WBE is a crucial aspect of understanding and managing the spread of infections across different areas. This approach involves mapping and analyzing the geographic distribution of viral loads in wastewater to gain insights into how infections spread across diverse regions and communities. By visualizing the spatial patterns of the viral presence and its concentrations, public health officials can identify hotspots, track the progression of infectious diseases, and implement targeted interventions to mitigate their impact. The use of GIS in particular plays an essential role in this process by providing the tools necessary to visualize and analyze wastewater data in relation to geographic features and populations. Employing GIS enables the integration of wastewater surveillance data with other spatially referenced information, such as population density, healthcare facility identification, and environmental characteristics. This integration allows for the creation of detailed maps that highlight areas with elevated viral loads and potential risk factors contributing to disease spread [2].

Using GIS, it is possible to overlay wastewater data with demographic information to assess the impact of socioeconomic factors on infection rates. For instance, by correlating viral concentrations in wastewater with areas of lower socioeconomic status, public health officials can identify communities that may require additional resources or targeted health campaigns to prevent outbreaks [3,20]. Furthermore, spatial analysis in WBE can be used to monitor the effectiveness of public health interventions over time and across different geographic scales. By tracking changes in viral loads in wastewater before and after interventions such as lockdowns, sanitation improvements, or vaccination drives, GIS can help evaluate their impact and guide adjustments to maximize their effectiveness [30,35].

There are several spatial analysis techniques that can be implemented in WBE for mapping and understanding the distribution and spread of viral loads across different regions. These techniques range from mapping and spatial clustering to geospatial predictive modeling and spatiotemporal dynamics, each providing unique insights into the behavior of viral infections in communities.

**Mapping Techniques**: Common mapping techniques like heat maps and choropleth maps are instrumental in representing viral concentrations across regions. Heat maps use color gradients to indicate the intensity of viral loads, highlighting areas with higher concentrations, while choropleth maps assign colors or patterns to predefined geographic areas based on the viral load values, enabling a clear visual differentiation between regions of varying viral presence. These visualizations help public health officials quickly identify areas of concern [3,63].

**Interpolative Methods**: Techniques like kriging and inverse distance weighting (IDW) are used to estimate viral loads in unsampled areas based on nearby sampled locations. By implementing the spatial autocorrelation principle, these methods enhance the comprehensiveness of spatial data, filling gaps and providing a more complete picture of viral distribution across the landscape. This approach is particularly useful in areas where direct sampling may be sparse or logistically challenging [2,20].

**Spatial Clustering Techniques**: Techniques such as Getis-Ord Gi* and Local Moran’s I identify clusters or hotspots of high viral loads. These analyses are critical in pinpointing areas that may require urgent intervention or targeted public health measures. By detecting clusters of high viral activity, officials can direct resources more efficiently and implement measures to prevent the spread of infections [30,35].

**Predictive Spatial Models**: These models use factors such as population density, sewer network characteristics, and environmental variables to predict the spread and concentration of viruses in wastewater. Integrating machine learning and spatial statistics, these models forecast future trends based on current and historical data, aiding in proactive public health planning. For instance, machine learning models can analyze patterns in viral load data alongside demographic information and environmental factors like temperature and rainfall, providing deeper insights into the dynamics of viral transmission and the impact of social determinants of health on the spread of infections [64,65].

**Spatiotemporal Analysis**: This analysis tracks changes in viral loads over time within specific geographic areas, revealing the evolution of outbreaks and the effectiveness of public health interventions. Time-series maps and animations visualize these dynamics, making the data accessible to both the public and policymakers. This visualization helps in understanding how the viral presence changes in response to interventions and can guide future public health strategies [18,19]. A short summary of several studies implementing different geospatial methods in wastewater surveillance is included in Table 2.

While some of the citations included in Table 2 pertain to bacteria, drugs, sewer networks, and anthropogenic variation sources, their relevance to virus studies within the context of WBE is significant. These studies provide essential insights into the methodologies and principles that underpin WBE. For instance, the detection and quantification techniques used for bacterial pathogens and pharmaceutical residues, such as PCR and NGS, are directly applicable to viral surveillance. Additionally, research on sewer networks and anthropogenic variations offers crucial information on sampling strategies, data interpretation, and the impacts of environmental factors on pathogen detection. By understanding the broader applications of WBE, we can better appreciate its robustness and adaptability for monitoring viral pathogens. This interdisciplinary approach enhances the comprehensiveness and effectiveness of WBE in regards to public health surveillance, particularly for tracking and managing viral outbreaks.

## 5. Enhancing Wastewater Surveillance with Advanced GIS Techniques

The analytical capabilities of GIS, including spatial interpolation, proximity analysis, and pattern recognition, deepen insights into wastewater data. Spatial interpolation estimates viral loads in unsampled areas, while proximity analysis links viral hotspots with essential infrastructure. Pattern recognition identifies trends and correlations between viral presence and various factors, enhancing surveillance accuracy [63].

The role of GIS in dynamic wastewater data monitoring enables real-time analysis and quick updates as new data arrive. This facilitates rapid responses to changes in viral concentrations, supporting timely public health interventions like local testing and quarantine measures [84]. Furthermore, GIS helps decision makers allocate resources more effectively by providing detailed spatial information on areas requiring interventions [85]. Analyzing wastewater data alongside population needs allows for the planning and optimizing of health services and infrastructure locations, ensuring that resources like testing facilities and vaccination centers are strategically placed for maximum impact [64,65,86].

### 5.1. Examples of GIS Enhancing Sampling Site Selection and Ensuring Equitable Monitoring

By analyzing geographic and demographic characteristics, GIS aids in determining the most optimal locations for wastewater sampling, taking into account factors like population density, accessibility to amenities, and the layout of the sewer network. Likewise, GIS is adept at identifying areas to ensure comprehensive coverage of a community, including both densely populated urban areas and more sparse rural zones. For example, in urban settings, GIS can pinpoint densely populated residential and commercial areas in which sampling might reveal a wealth of detailed public health data. Conversely, in rural areas, GIS helps in locating scattered populations connected by sparse sewer networks, ensuring that these communities are not overlooked in health surveillance efforts [2,20].

One critical function of GIS in WBE is identifying underserved or vulnerable populations that might be missed by traditional health surveillance methods. By overlaying socioeconomic data with wastewater flow patterns, GIS can highlight neighborhoods or communities that lack health resources or have higher susceptibility to diseases. This ensures that these populations are included in wastewater monitoring efforts, fostering a more equitable approach to public health [3,63,87].

Several case studies have demonstrated the impact of GIS-driven adjustments in sampling strategies. For instance, a project in a mid-sized city used GIS to analyze the sewer network and demographic data, leading to the redistribution of sampling sites. This adjustment uncovered hidden outbreaks in marginalized neighborhoods, guiding targeted public health interventions like increased testing and community-specific health campaigns [29,35]. GIS data also supports the efficient allocation of resources by pinpointing areas with higher viral loads or increased risk of transmission. This precision allows public health officials to direct interventions such as sanitation upgrades, vaccination drives, or health education campaigns precisely where they are most needed, maximizing the impact of these efforts [18,19,87,88]. Moreover, GIS aids in planning the expansion or modification of wastewater infrastructure to enhance surveillance capabilities and response readiness. By analyzing current and projected data, GIS can recommend where additional sampling points or infrastructure enhancements are necessary to improve coverage and sensitivity in detecting viral spread [89,90].

Furthermore, the integration of GIS with remote sensing data represents a significant advancement in spatial analysis, particularly in the context of WBE. This combination enhances the capability to monitor and predict viral transmission and persistence by incorporating a broader range of environmental variables.

**GIS and Remote Sensing Synergy**: GIS provides a robust framework for analyzing and visualizing spatial data, while remote sensing offers a means to acquire environmental and geographic information from a distance. Together, they enrich wastewater surveillance by integrating environmental variables that directly or indirectly affect viral transmission and persistence. Remote sensing data, such as satellite imagery, aerial photography, and drone-captured data, contribute critical insights regarding land use, vegetation cover, temperature, and water bodies [91,92]. These insights are crucial for understanding how environmental factors influence viral concentrations and movements in wastewater.

**Environmental Correlates and Viral Dynamics**: GIS and remote sensing are used to map environmental correlates like temperature, humidity, and precipitation, which are known to impact virus survival and transmission [93,94]. For example, mapping these factors can help predict potential hotspots or explain fluctuations in viral loads in wastewater based on changing environmental conditions. Such mapping has been critical in areas where seasonal changes significantly affect viral persistence, allowing public health officials to anticipate and prepare for potential outbreaks.

**Real-Time Data and Adaptive Management**: The capability of remote sensing to provide real-time or near-real-time environmental data, combined with GIS, creates the possibility of making dynamic adjustments in monitoring and response strategies as conditions change. This approach supports adaptive management in public health, allowing for quick shifts in surveillance focus or intervention strategies based on the latest data. It ensures that resources and efforts are directed efficiently and effectively based on current environmental and epidemiological conditions [30,35].

**Predictive Modeling in Public Health and WBE**: As mentioned in a previous section, predictive modeling uses historical and current data to forecast future trends in virus spread. GIS enhances these models by providing a spatial framework that allows for the mapping and analysis of data over geographic regions, adding vital context to predictions. These models are used in scenario planning to assess the potential impact of different public health interventions and environmental changes on virus spread. They help evaluate the risk of spread in various scenarios, supporting decision makers in choosing optimal strategies for containment and mitigation [95,96].

**Real-Time Responses and Resource Allocation**: The combination of predictive modeling and GIS supports real-time public health responses by providing actionable forecasts that can be visualized on maps and shared with decision makers and the public [97]. This integration facilitates the allocation of resources, such as testing, vaccination, and sanitation efforts, to areas predicted to experience higher viral loads. For instance, in a study where predictive modeling combined with GIS forecast virus spread patterns, public health policies were adjusted, and interventions were successfully implemented, leading to a predictive reduction in transmission rates [98].

Real-world examples where predictive modeling combined with GIS has successfully forecast virus spread patterns include tracking seasonal influenza and monitoring the re-emergence of norovirus in communities [99,100]. These applications have influenced public health policies, leading to successful interventions. The lessons learned from these cases highlight the importance of integrating environmental data with epidemiological insights, ensuring that responses are timely, targeted, and effective in mitigating public health risks [2,20].

Other examples included the surveillance of viruses like hepatitis A, norovirus, and SARS-CoV-2. In Italy, GIS was used to monitor the hepatitis A virus in urban sewage, correlating environmental data with clinical cases during an outbreak in 2013. The use of GIS allowed health officials to analyze data by region and catchment area, considering only cases reported by those local health units corresponding to the geographical areas served by the participating wastewater treatment plants. This approach enabled a more targeted and effective response to the outbreak, demonstrating the power of GIS in mapping and responding to viral threats in real time [101]. Likewise, a study explored the cross-correlations between human norovirus GII wastewater data and various traditional epidemiological surveillance data, i.e., syndromic, outbreak, and Google search term data [102]. The study found that norovirus-specific measures and earlier detection in specific locations using wastewater surveillance could significantly enhance public health surveillance efforts. This approach showed the potential of wastewater-based surveillance for providing localized, early-warning information to inform public health decision making, especially for viruses like norovirus, which often go undetected due to asymptomatic cases. 

More recently, at the University of California San Diego, GIS was crucial in detecting COVID-19 in wastewater before local individuals showed symptoms [103]. The university connected its wastewater testing results to a live GIS map, showing real-time positive COVID-19 readings in specific campus buildings. This innovative use of GIS enabled swift communication and encouraged testing among at-risk individuals, effectively constraining the virus’s spread on campus. Likewise, in Denmark, wastewater surveillance for SARS-CoV-2 was implemented across 230 locations, covering over 85% of the population [104]. This extensive network allowed for the early detection of local outbreaks in low-prevalence situations. By combining GIS data of the locations and the Monte Carlo simulation model, the study explored the feasibility of setting up an early warning system based on wastewater data, highlighting the potential for proactive local interventions to prevent wider virus spread.

### 5.2. Challenges and Limitations 

The implementation of GIS in WBE exhibits challenges such as ensuring data privacy, maintaining consistent data quality, and integrating diverse data sources. Privacy concerns require transparent data collection, analysis, and sharing practices, along with robust legal frameworks to prevent stigmatization or discrimination [20,29,105]. Maintaining data quality in WBE involves standardizing protocols across laboratories and regions and addressing environmental factors, e.g., temperature and precipitation, that affect viral detection [2,3].

Integrating viral load with demographic, environmental, and clinical data in a GIS environment poses technical and logistical challenges. Addressing issues of data compatibility, scale, and resolution necessitates sophisticated data management and governance [106,107]. Additionally, effective wastewater surveillance demands advanced laboratory equipment, skilled personnel, and consistent funding, particularly to overcome disparities between high-income and low-income regions and to enhance global health security through training and technology transfer [18,19].

Lastly, collaboration with policymakers, public health officials, community leaders, and communication experts is essential to translate surveillance findings into actionable public health strategies and interventions, ensuring that information is effectively communicated and that privacy concerns are addressed [108,109]. Overall, addressing these challenges through collaboration, capacity building, and advanced technologies is crucial for the success and sustainability of WBE efforts in monitoring and responding to public health threats.

## 6. Future Directions and Recommendations

WBE is evolving with the introduction of more sensitive and rapid pathogen detection methods like NGS, CRISPR technologies, and biosensors. These advancements enable real-time, accurate detection of a broad spectrum of viral and bacterial pathogens in wastewater, often enhanced by portable or on-site testing equipment for immediate monitoring and decision making [20,110,111]. Moreover, technological advancements also improve the spatial and temporal resolution of the monitoring, allowing for the precise mapping of virus spread and early detection of hotspots. The integration of drones, satellite imagery, and GIS enhances environmental monitoring, providing comprehensive data integration from remote sensing with wastewater surveillance [3,63,112,113]. Automated systems further revolutionize WBE by streamlining sample collection, processing, and analysis, reducing human error and labor costs, and ensuring consistent data quality [30,35].

The integration of wastewater surveillance with other health monitoring data offers a holistic view of public health, enabling comprehensive risk assessment and targeted interventions. Advancements in bioinformatics and metagenomics allow for the identification of both known and emerging pathogens, enriching our understanding of public health threats [18,41,114,115]. Developing global databases and collaborative platforms is essential for international cooperation, enhancing global health security by exchanging methodologies and insights across regions [2,116]. Integrating wastewater surveillance into the public health framework, combined with clinical and epidemiological data, aids in informed policymaking and intervention planning. Dedicated teams within health departments are necessary to ensure continuity and expertise in wastewater surveillance [2,117].

Other important approaches for WBE implementation include the monitoring of emerging pathogens and zoonotic diseases, a practice which would significantly bolster early detection capabilities. When combined with developing protocols for detecting a wider range of pathogens, including those with the potential to jump from animals to humans, WBE can become a more comprehensive surveillance tool, ready to address future pandemics. Likewise, incorporating machine learning and artificial intelligence (AI) algorithms to analyze complex wastewater datasets can enhance predictive modeling and trend analysis. These advanced technologies can help identify patterns and correlations that are not immediately apparent, providing more accurate predictions of disease outbreaks and their potential impacts on communities. This technological integration can lead to more precise and timely public health interventions.

Fostering interdisciplinary collaborations between epidemiologists, environmental scientists, data analysts, and public health officials is essential for the holistic development of WBE. Such collaborations can lead to the creation of integrated surveillance frameworks that combine insights from various fields, improving the overall effectiveness and responsiveness of public health strategies. By working together, these experts can develop innovative solutions and methodologies that enhance the utility of WBE. Moreover, establishing community engagement and education programs to raise awareness about the importance and benefits of WBE can enhance public support of and participation in these programs. Educating the public on how wastewater surveillance contributes to health security can help engender community trust and cooperation, which is crucial for the successful implementation of surveillance programs. Public awareness initiatives can ensure that communities understand the value of their participation in these surveillance efforts.

Exploring the use of blockchain technology for secure and transparent data sharing among different stakeholders can address privacy and data integrity concerns. Blockchain technology can ensure that sensitive health data collected through WBE is securely shared and accessed only by authorized entities, maintaining confidentiality while enabling comprehensive analysis. This secure data management approach can foster greater collaboration and trust among stakeholders. Furthermore, integrating WBE with smart city initiatives can create more resilient urban health monitoring systems. Smart cities equipped with advanced sensors and IoT (Internet of things) devices can provide real-time data for various environmental and health parameters. When combined with WBE data, this can offer a detailed and dynamic picture of public health trends, enhancing the ability of cities to swiftly and effectively respond to health threats. 

Investment in research and development is needed to enhance the sensitivity, speed, and scalability of wastewater surveillance. Pilot programs and studies exploring applications like tracking antimicrobial resistance or monitoring non-infectious health indicators are vital for expanding the scope and impact of WBE [22,118,119]. Policymakers must recognize and support wastewater surveillance as a key public health strategy, developing regulatory frameworks that protect privacy and ethical considerations while enabling effective data use. Including wastewater surveillance in long-term public health planning is crucial for a resilient health system [2,40]. Furthermore, advocating for the standardization of WBE methods and protocols at the international level can facilitate more effective comparisons and collaborations between countries. International standards can help harmonize data collection, analysis, and reporting processes, enhancing the global utility of wastewater surveillance data in managing cross-border health threats. Standardization efforts can ensure that WBE contributes effectively to global health security, providing consistent and reliable data for public health decision making worldwide. Embracing these innovations ensures that wastewater surveillance significantly enhances public health responses, preparedness, and resilience.

## Figures and Tables

**Table 1 pathogens-13-00685-t001:** Modeling techniques and practical applications in wastewater-based epidemiology.

Model Types	Model Names	Spatiotemporal Scales	References	Key Findings
Statistical	Time series model	Point-based; daily	[18]	The epidemiological time series model can estimate changes in COVID-19 prevalence based on the measurement of SARS-CoV-2 RNA concentrations in primary sludge.
Statistical	Mass balance; Monte Carlo simulation	Point-based; daily	[20]	The number of infected individuals is simulated using a Monte Carlo approach based on a mass balance of the number of viral RNA copies in wastewater.
Statistical	Viral dynamics model; Markov chain Monte Carlo simulation	Neighborhood level; daily	[23]	The wastewater viral load is modeled to investigate viral shedding dynamics. The findings suggest that SARS-CoV-2 concentration in wastewater is closely related to newly infected cases.
Process-based	SWMM	Subcatchment; hourly	[34]	A hydrologic–hydraulic model, SWMM, is used to simulate the wastewater flow rates, velocities, and in-sewer travel time for tracking SARS-CoV-2/COVID-19 at epidemic hotspots.
Data-driven	Artificial neural network	Case-based; daily	[46]	A data-driven model using an artificial neural network is developed to estimate the COVID-19 prevalence rates based on wastewater data and other influencing factors.
Data-driven	Multiple linear regression; artificial neural network; adaptive neuro fuzzy inference system	Case-based; daily	[55]	Three different data-driven models are used for COVID-19 prevalence prediction, namely MLR, ANN, and ANFIS. ANN and ANFIS show better predication capability than MLR, and ANN shows stronger robustness than ANFIS.
Metapopulation	Mobility network model	Point-based; hourly	[56]	A metapopulation mobility network model is developed in 10 large US metropolitan areas to simulate the spread of SARS-CoV-2. The modeling results can guide policy-makers to generate better reopening approaches.
Metapopulation	Mobility network model	NUTS3 administrative unit level; weekly	[57]	Using a meta population model of COVID-19 transmission, the study finds that appropriate community coordination is important for easing nonpharmaceutical interventions, preventing resurgence and spread.

**Table 2 pathogens-13-00685-t002:** Methods for geospatial analysis with examples of applications and findings.

Methods	Geospatial Techniques	Geographic Unit and Location	Reference	Key Findings
**Mapping techniques**	Data aggregation to district level and GIS mapping	District level in three counties in northern New York State, USA	[66]	A comprehensive database was created for all municipal sewer sheds in New York State to support statewide wastewater surveillance. This involved combining public tax records, sewer access information, physical maps, and municipal records to create digital boundaries compatible with GIS.
	GIS methodology of spatial joining	Electoral wards in Greater Sydney, Australia	[67]	The spatial join tool was employed to overlay these data layers onto the wastewater network layer. This spatial join provided a detailed geospatial analysis, enabling the identification of trends and correlations between the presence of pathogens and various demographic, socioeconomic, and healthcare-related factors within the Sydney urban area. Endemicity of ESBL-E isolates was identified, while CRE, VRE, and MRSA were detected sporadically. ESBL-E load correlated with demographic factors, CRE load was linked to hospital stays, and VRE load was associated with the number of schools, but not with healthcare facilities.
	Data aggregation to electoral wards and GIS mapping	Electoral wards in China	[68]	Significant regional differences in caffeine consumption were found, with the highest occurring in East China and the lowest in Northeast China. Caffeine consumption correlated with GDP and urban residents’ disposable income, and there was a notable correlation between caffeine and cotinine concentrations.
	Data aggregation to county level and GIS mapping	County level in California, Florida, Iowa, Maryland, Michigan, Minnesota, New Jersey, New York, and the U.S. island areas.	[45]	U.S. Census data revealed lower-than-average sewer connectivity for certain demographic groups. Geographic areas with low sewer connectivity were identified, including Alaska, the Navajo Nation, and parts of Minnesota, Michigan, and Florida.
	Data aggregation to electoral wards and GIS mapping GIS methodology of spatial joining	Electoral wards in Cape Town, South Africa	[69]	The highest viral RNA signal was observed in the first week, with a decline over six weeks corresponding to decreasing COVID-19 cases. An early warning system for future waves was established.
**Interpolative methods**	Inverse distance weighted (IDW) interpolation	Electoral wards in Taoyuan City, Taiwan	[70]	Wastewater viral concentrations correlated with COVID-19 case numbers. Several treatment technologies effectively eliminated viral RNA from wastewater influent. The IDW interpolation and hotspot model using GIS analyzed spatiotemporal variations of SARS-CoV-2 in wastewater.
**Spatial clustering**	Local Moran’s I and the Getis-Ord Gi*	Neighborhood level in Reno–Sparks metropolitan area	[71]	Local Moran’s I and Getis-Ord Gi* were utilized to analyze the spatial distribution of SARS-CoV-2 viral RNA concentrations in wastewater. Local Moran’s I identified spatial clustering patterns, detecting high–high clusters and outliers within these clusters, while Getis-Ord Gi* pinpointed hot and cold spots, revealing significant areas of high and low viral concentrations. Distinct time series patterns were identified among different sewer sheds. Demographic parameters such as population density, poverty levels, household income, and age showed important gradients across these areas.
	K-means clustering	Electoral wards in coastal aquifer	[72]	K-means clustering based on PCA identified three hydrogeochemical classes and their sources, delineating natural and anthropogenic variations in the aquifer.
	Getis-Ord Gi*	Individual level in Rawalpindi, Pakistan	[20]	Malaria cases were mainly related to agriculture, low vegetation, and water classes. Temporal variation of malaria cases showed a significant positive association with average monthly rainfall and temperature.
	Localized hotspots	Electoral wards in Southeastern Virginia, USA	[73]	SARS-CoV-2 detections were initially sporadic, increasing in frequency and concentration from mid-March to late July. Population-normalized viral load fluctuations in several WWTP service areas corresponded with known outbreaks.
**Predictive spatial models**	Spatiotemporal autoregressive structure model	Census-tract level in England	[74]	The model predicted SARS-CoV-2 viral concentration in wastewater at high spatiotemporal resolution, serving as an early warning tool for public health surveillance.
	Bayesian spatial analysis	County level in Louisville, KY, USA	[75]	Including wastewater data slightly improved nowcast accuracy compared to models using case and death data alone, highlighting the value of incorporating wastewater data for enhanced public health surveillance.
	Hydrologically and hydraulically influenced spatial statistical approach	Census-tract level in Hamilton County, OH, USA	[76]	The study predicted a 90% loss of fecal information from a given census block group over a travel time of 10.3 h.
	Bayesian hierarchical model	Lower tier local authority level in England	[77]	Wastewater data improved nowcast accuracy and reduced uncertainty, emphasizing the importance of maintaining prevalence data at a national level during non-epidemic periods.
**Temporal-Spatial Analysis**	Spatiotemporal analysis and modeling	Electoral wards in University of North Carolina at Charlotte, NC, USA	[78]	A web-based spatial decision support system-facilitated efficient management, analytics, and sharing of wastewater testing data, aiding in COVID-19 prevention and mitigation on campus.
	Space–time analysis	The locations of structures in the administrative unit in Austin, TX, USA	[79]	Variations in lag-times between wastewater loading and case reports were observed, with daily case reports in some locations closely following viral load trends. Socio-demographic characteristics influenced these findings.
	Metagenomic analysis	Electoral wards in Spain	[80]	Metagenomic analysis of SARS-CoV-2 in wastewater allowed for the detection of mutations defining the B.1.1.7 lineage and anticipation of certain mutations before clinical detection.
	Spatial and temporal analysis	Individual level in San Francisco Bay Area, USA	[81]	CrAssphages displayed minimal spatial and temporal variability. Significant correlations were found between unnormalized SARS-CoV-2 RNA signals and clinical testing data, with locational dependencies impacting the lead time for clinical trends.
	Bayesian stochastic search variable selection (BSSVS) procedure	Electoral wards in Buenos Aires, Argentina	[82]	Spatiotemporal diffusion analysis suggested Europe as an intermediate path for NoV dissemination between the Americas and the rest of the world, emphasizing the public health risks from sewage discharges.
**Integration of Spatial Analysis and GIS**	Data aggregation to district level and GIS mapping	District level in three counties in northern New York State, USA	[66]	A statewide wastewater surveillance database for New York State was created using public tax records, sewer access information, physical maps, and municipal records to draw GIS-compatible digital boundaries.
**Real time responses and resource allocation**	Optimal site selection in GIS	Individual level in NE Iberian Peninsula, Girona, Spain	[83]	Algorithms proposed optimal station locations, covering significant manhole areas and a high percentage of inhabitants, ensuring efficient resource allocation.

## Data Availability

Not applicable.

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
