# Peer review of "Advancing Public Health Surveillance: Integrating Modeling and GIS in the Wastewater-Based Epidemiology of Viruses, a Narrative Review"

_pathogens, 2024, doi:10.3390/pathogens13080685_

Round 1
Reviewer 1 Report
Comments and Suggestions for Authors
The authors prepared a review paper on wastewater-based epidemiology (WBE), which increases its significance in public health after COVID-19 pandemic. The manuscript can be published after minor revisions as follows.
1) Use Italic font for “Salmonella” and “Escherichia coli”.
2) The caption of Table 1 (Table 1. Modeling methods for wastewater-based epidemiology with example applications) gives strange impression. Consider again especially for “example applications”.
3) The subheading “3. Modeling Techniques in Wastewater Surveillance” may not be appropriate, because “4. Spatial Analysis in Wastewater Surveillance” also deals modeling. Do the authors mean “modeling time-dependent changes in wastewater surveillance”?
4) In table 2, the writing style for “key findings” is not uniform. For example, the authors use “We” as to the references [35][46] and [35] in table 2. This is not usual as a style for the introduction of researches in this table. In addition, some of the “key findings” do not provide generalized findings. For example, the key findings of [53] include “Deploying 5, 6 or 7 stations results in more than 80% coverage in manholes and more than 85% of the inhabitants”, which has no generalized significance (key findings). Another problem is the syntax errors of the descriptions for some of the key findings. The authors are advised to check the key findings in this table again to show more general findings with a correct and unified way.
5) The difference is not clear between “4. Spatial Analysis in Wastewater Surveillance” and “5. Integration of Spatial Analysis and GIS in Wastewater Surveillance”. Consider again the chapters in this review article.
Comments on the Quality of English LanguageThe reviewer did not find fatal problems in the language.
Author Response
Reviewer #1
The authors prepared a review paper on wastewater-based epidemiology (WBE), which increases its significance in public health after COVID-19 pandemic. The manuscript can be published after minor revisions as follows.
We thank the reviewer for the positive feedback provided to our manuscript. We have addressed below the comments from the review
1) Use Italic font for “Salmonella” and “Escherichia coli”.
We thank the review for noticing this. We have made the change as suggested by the reviewer in the revised version of the document (line 66 in the marked document)
2) The caption of Table 1 (Table 1. Modeling methods for wastewater-based epidemiology with example applications) gives strange impression. Consider again especially for “example applications”.
We had changed the tittle of the table as suggested by the reviewer (lines 308 and 309 in the marked document)
3) The subheading “3. Modeling Techniques in Wastewater Surveillance” may not be appropriate, because “4. Spatial Analysis in Wastewater Surveillance” also deals modeling. Do the authors mean “modeling time-dependent changes in wastewater surveillance”?
We have changed the subheading as suggested by the reviewer (line 258 in the marked document)
4) In table 2, the writing style for “key findings” is not uniform. For example, the authors use “We” as to the references [35][46] and [35] in table 2. This is not usual as a style for the introduction of researches in this table. In addition, some of the “key findings” do not provide generalized findings. For example, the key findings of [53] include “Deploying 5, 6 or 7 stations results in more than 80% coverage in manholes and more than 85% of the inhabitants”, which has no generalized significance (key findings). Another problem is the syntax errors of the descriptions for some of the key findings. The authors are advised to check the key findings in this table again to show more general findings with a correct and unified way.
We thank the reviewer for their detailed and constructive feedback regarding Table 2. We have thoroughly revised the "Key Findings" column to ensure a uniform writing style, removing any first-person references such as “We.” Additionally, we have generalized the findings to enhance their broader significance and corrected any syntax errors. These improvements aim to provide clearer, more generalized insights from each study, presented in a consistent and accurate manner.
5) The difference is not clear between “4. Spatial Analysis in Wastewater Surveillance” and “5. Integration of Spatial Analysis and GIS in Wastewater Surveillance”. Consider again the chapters in this review article.
We appreciate the reviewer's insightful comment regarding the clarity between sections 4 and 5. To address this, we have renamed section 5 to "Enhancing Wastewater Surveillance with Advanced GIS Techniques." This revised heading better reflects the distinct focus of this section, which now specifically addresses the integration of advanced GIS tools and techniques in wastewater surveillance, beyond the basic spatial analysis methods discussed in section 4. This change emphasizes the added value of sophisticated GIS applications in improving real-time monitoring, resource allocation, and overall public health decision-making.
Reviewer 2 Report
Comments and Suggestions for Authors
This review article presents a comprehensive examination of the use of modeling, spatial analysis, and Geographic Information Systems (GIS) in the surveillance of viruses in wastewater. In addition, it explores the application of various modeling techniques that enable the prediction and understanding of virus concentrations and spread patterns in wastewater system. The review is well written but needs to consider the following comments before any further recommendations.
1- In such reviews, the presence of some figures in addition to the tables is very important. For this reason, some figures could be added to give more impressive information about any idea.
2- The mathematical expression of any model used should be added in order to simplify its understanding. The authors mentioned many type of models without giving any mathematical expression about them.
3- Some researches concerning COVID-19 are missed in the literature list. I think their addition and discussion will be an added value to the review such as:
a- Milad Mousazadeh, Zohreh Naghdali, Neda Rahimian, Marjan Hashemi, Biswaranjan Paital, Zakaria Al-Qodah, Ahmad Mukhtar, Rama Rao Karri, Alaa El Din Mahmoud, Mika Sillanpää, Mohammad Hadi Dehghani, Mohammad Mahdi Emamjomeh, Management of environmental health to prevent an outbreak of COVID-19: a review, Environmental and health management of novel coronavirus disease (COVID-19, Academic Press, 2021, 235-267.
4- The recommendations for future work is an excellent addition in this review. I advise to add more recommendations.
Author Response
Reviewer # 2
This review article presents a comprehensive examination of the use of modeling, spatial analysis, and Geographic Information Systems (GIS) in the surveillance of viruses in wastewater. In addition, it explores the application of various modeling techniques that enable the prediction and understanding of virus concentrations and spread patterns in wastewater system. The review is well written but needs to consider the following comments before any further recommendations.
We thank the reviewer for their positive feedback and appreciation of our review article. We are grateful for the constructive comments provided and have addressed them to enhance the quality and clarity of our manuscript. We believe these revisions will further strengthen our examination of modeling, spatial analysis, and GIS in wastewater-based epidemiology.
1- In such reviews, the presence of some figures in addition to the tables is very important. For this reason, some figures could be added to give more impressive information about any idea.
We appreciate the reviewer's suggestion regarding the inclusion of figures to enhance the presentation of information. While we acknowledge that figures can be valuable in illustrating concepts, we believe that the content and the comprehensive tables provided in our review effectively summarize the landscape of Wastewater Surveillance and GIS. These tables are designed to offer a clear and concise overview of the key methodologies and findings, achieving the goal of the paper. Therefore, we do not believe that adding figures would substantially contribute to the clarity or impact of the manuscript at this time. We thank the reviewer for their understanding and consideration.
2- The mathematical expression of any model used should be added in order to simplify its understanding. The authors mentioned many type of models without giving any mathematical expression about them.
We appreciate the reviewer's suggestion regarding the inclusion of mathematical expressions to enhance the understanding of the models discussed. While we acknowledge the importance of mathematical equations in elucidating model details, we believe that their inclusion is not necessary for the message we aim to deliver in this narrative review. The purpose of this manuscript is to provide an overview and synthesis of the use of modeling and GIS in wastewater surveillance, rather than a detailed technical exposition of specific models. Including mathematical expressions would shift the focus and potentially complicate the narrative. Therefore, we have opted to describe the models conceptually to maintain clarity and accessibility for a broader audience. Thank you for your understanding.
3- Some researches concerning COVID-19 are missed in the literature list. I think their addition and discussion will be an added value to the review such as: a- Milad Mousazadeh, Zohreh Naghdali, Neda Rahimian, Marjan Hashemi, Biswaranjan Paital, Zakaria Al-Qodah, Ahmad Mukhtar, Rama Rao Karri, Alaa El Din Mahmoud, Mika Sillanpää, Mohammad Hadi Dehghani, Mohammad Mahdi Emamjomeh, Management of environmental health to prevent an outbreak of COVID-19: a review, Environmental and health management of novel coronavirus disease (COVID-19, Academic Press, 2021, 235-267.
We appreciate the reviewer's suggestion and have added several key studies to our literature review. We included the suggested references and other references like Prado et al. (2021) that highlighted the correlation between viral loads in wastewater and COVID-19 case numbers, demonstrating WBE's potential as an early warning system. Gonçalves et al. (2022) reviewed centralized and decentralized WBE approaches, emphasizing their utility for comprehensive surveillance and targeted interventions. Medema et al. (2020) and Ahmed et al. (2020) provided evidence from the Netherlands and Australia, respectively, supporting the use of WBE to monitor COVID-19 prevalence. Ciannella et al. (2023) conducted a systematic review, confirming the effectiveness of WBE in supplementing clinical testing despite methodological challenges. By adding and discussing these studies, we believe our review now provides a more comprehensive overview of the current state and potential of WBE in managing the COVID-19 pandemic. Thank you for your valuable feedback, which has enhanced the depth and relevance of our manuscript. (lines 115 to 160 in the marked document)
4- The recommendations for future work is an excellent addition in this review. I advise to add more recommendations.
We thank the reviewer for the positive comment. We have followed the reviewer’s suggestion and expanded this section of the manuscript to include more recommendations. (lines 585 to 618 in the marked document)
Reviewer 3 Report
Comments and Suggestions for Authors
Wastewater-based epidemiology of viruses has gained significant recognition and application since the COVID-19 pandemic. This review discusses the integration of modelling and GIS in WBE, a topic likely to interest many readers. I agree with the authors that statistical modelling and GIS can greatly enhance virus monitoring in wastewater. However, several issues need to be addressed in the manuscript.
The main concerns are the cited articles do not match the claims in the manuscript. For examples,
Line 152: The discussion on “outbreak prediction based on factors like weather and water quality” cites reference 25, which actually focuses on illicit drug use in 19 European cities through sewage analysis.
Lines 378-383: The claim that “several case studies have demonstrated the impact of GIS-driven adjustments ... guiding targeted public health interventions like increased testing and community-specific health campaigns” is not supported by references 9 and 18, as they do not employ GIS.
Lines 402-404 and 431-434: The cited works do not match the statements made.
These are just several examples, and there are more. These citation errors undermine the manuscript's credibility, a critical flaw in a review article.
Additional Issues:
1. There are many repetitions in the writing, for example,
Introduction and Part 2 (Lines 126-142): Both sections discuss the benefits of WBE, leading to redundancy.
Introductions of Parts 4 and 5: Both sections redundantly highlight the benefits of GIS in WBE.
2. Lines 111-112: The authors claim, “particularly in the development of standardized protocols for the detection of SARS-CoV-2.” However, no standardized protocols exist as sampling, virus concentration, and detection methods vary between labs.
3. Part 3: While several modelling methods for WBE are listed, it would be beneficial to discuss the advantages and disadvantages of each model and their applications in different scenarios.
4. Table 2: Many citations pertain to bacteria, drugs, sewer networks, and anthropogenic variation sources rather than viruses. The relevance of these works to virus studies should be clarified.
5. Key Findings in Table 2: The key findings often directly quote the abstracts of the cited works without summarizing, causing confusion. For example, reference 53’s key finding is an extract that lacks context for the reader. The reader do not know what “Deploying 5, 6 or 7 stations” means.
6. Some references, such as 17 and 19, are incomplete and need verification.
Author Response
Reviewer # 3
Wastewater-based epidemiology of viruses has gained significant recognition and application since the COVID-19 pandemic. This review discusses the integration of modelling and GIS in WBE, a topic likely to interest many readers. I agree with the authors that statistical modelling and GIS can greatly enhance virus monitoring in wastewater. However, several issues need to be addressed in the manuscript.
We thank the reviewer for their positive feedback and recognition of the significance of our review topic. We appreciate your agreement on the importance of integrating statistical modeling and GIS in enhancing virus monitoring in wastewater. We have carefully considered your comments and addressed the issues raised to improve the quality and comprehensiveness of our manuscript.
The main concerns are the cited articles do not match the claims in the manuscript. For examples,
Line 152: The discussion on “outbreak prediction based on factors like weather and water quality” cites reference 25, which actually focuses on illicit drug use in 19 European cities through sewage analysis.
Lines 378-383: The claim that “several case studies have demonstrated the impact of GIS-driven adjustments ... guiding targeted public health interventions like increased testing and community-specific health campaigns” is not supported by references 9 and 18, as they do not employ GIS.
Lines 402-404 and 431-434: The cited works do not match the statements made.
These are just several examples, and there are more. These citation errors undermine the manuscript's credibility, a critical flaw in a review article.
We thank the reviewer for their meticulous review and for pointing out the inconsistencies between the cited articles and the claims made in our manuscript. The issues arose due to problems with our citation manager software. We have now thoroughly reviewed and corrected all citations to ensure they accurately support the statements made in the manuscript. We have fixed the issue and updated the references, ensuring they directly correlate with the claims discussed.
Additional Issues:
- There are many repetitions in the writing, for example,
Introduction and Part 2 (Lines 126-142): Both sections discuss the benefits of WBE, leading to redundancy.
Introductions of Parts 4 and 5: Both sections redundantly highlight the benefits of GIS in WBE.
We thank the reviewer for highlighting the repetitions in our manuscript. We have addressed this issue by streamlining the content to remove redundant discussions. Specifically, we have combined the overlapping sections in the Introduction to provide a unified discussion on the benefits of WBE and to eliminate repetitive content regarding the benefits of GIS in WBE, ensuring that each section presents distinct and complementary information. These changes enhance the clarity and coherence of our manuscript. (pages 2 and 3 in the marked document)
- Lines 111-112: The authors claim, “particularly in the development of standardized protocols for the detection of SARS-CoV-2.” However, no standardized protocols exist as sampling, virus concentration, and detection methods vary between labs.
We appreciate the reviewer's valuable comment regarding the variability in protocols for the detection of SARS-CoV-2. We acknowledge that, currently, there is no universally accepted standardized protocol for sampling, virus concentration, and detection methods, which indeed vary between laboratories. In response, we have revised the manuscript to reflect this reality and emphasize the ongoing efforts and the need for developing such standardized protocols. This change highlights the importance of harmonizing methodologies to improve comparability and reliability of wastewater-based epidemiology (WBE) data globally. We hope this revision clarifies our position and aligns with the reviewer's observations. (lines 115 to 122 in the marked document)
- Part 3: While several modelling methods for WBE are listed, it would be beneficial to discuss the advantages and disadvantages of each model and their applications in different scenarios.
We appreciate the reviewer's suggestion to discuss the advantages and disadvantages of each modeling method in greater detail. While we agree that such an analysis would be valuable, we believe that the current description of the models and the tables summarizing some modeling studies are sufficient for the scope of our review. Our primary focus is on providing a comprehensive overview of the use of WBE and GIS in public health surveillance, rather than an in-depth exploration of modeling techniques. A more detailed examination of the modeling aspects would be beyond the intended scope of this review and might be better suited for a separate review focusing specifically on modeling in WBE. Thank you for your understanding.
- Table 2: Many citations pertain to bacteria, drugs, sewer networks, and anthropogenic variation sources rather than viruses. The relevance of these works to virus studies should be clarified.
We appreciate the reviewer's observation regarding the citations that pertain to bacteria, drugs, sewer networks, and anthropogenic variation sources. We included these references to illustrate the broader context and versatility of wastewater-based epidemiology (WBE) in monitoring various public health indicators. The methodologies and findings from these studies provide foundational knowledge and demonstrate the utility of WBE, which is directly applicable to virus detection and surveillance. We have revised the manuscript to clarify the relevance of these works to virus studies, emphasizing how the principles and techniques used in these broader applications contribute to the robustness and adaptability of WBE for viral monitoring. This context helps to underscore the interdisciplinary nature of WBE and its comprehensive potential in public health surveillance. (lines 395 to 406 in the marked document)
- Key Findings in Table 2: The key findings often directly quote the abstracts of the cited works without summarizing, causing confusion. For example, reference 53’s key finding is an extract that lacks context for the reader. The reader do not know what “Deploying 5, 6 or 7 stations” means.
We thank the reviewer for pointing out the issues with the Key Findings in Table 2. We have revised the table to improve the summaries of the studies described. We have contextualized the key findings from the studies included in the table to provide clear and concise summaries that enhance the reader's understanding. These improvements ensure that the key findings are accurately represented and easily comprehensible.
- Some references, such as 17 and 19, are incomplete and need verification.
As it was mentioned in a previous comment, we have addressed the issue with citations in the revised version of the manuscript.
Round 2
Reviewer 3 Report
Comments and Suggestions for Authors
The authors have corrected the errors in reference citations and addressed most of the concerns in this version of the manuscript. However, a few points still need to be corrected or clarified before acceptance for publication:
1. In Table 2: Reference 67 appears twice, and Reference 71 appears three times.
For Reference 67, it is stated that two geospatial techniques are used: “GIS methodology of spatial join” and “Forward and backward stepwise model”. However, the cited publication states, “The best regression models were selected using the forward and backward stepwise model selection method using the step function in the R-studio software.” Therefore, the “Forward and backward stepwise model” is more likely a statistical method instead of a geospatial technique.
For Reference 71, three geospatial techniques are listed: “Local Moran's I and Getis-Ord Gi*”, “Spatial sampling strategy, Local spatial autocorrelation”, and “spatial sampling strategy across neighborhood-scale sewershed catchments”. Clarification is needed on whether points 2 and 3 are the same. Additionally, in the cited publication, “Local Moran's I and Getis-Ord Gi*” seem to fall within the Local spatial autocorrelation methodology. The authors should clarify these issues.
2. In the revised manuscript, Lines 112-114 contain the newly added content, “…… for optimal indoor air quality……”, which seems irrelevant to the other content on WBE in this paragraph.
3. The reference citations have been improved but still show some inconsistencies in format. For example, Lines 55, 74, 114, 132 and more (citations appear before punctuation in some places and after punctuation in others). These small detail errors should be resolved before acceptance.
Author Response
Reviewer #1
- In Table 2: Reference 67 appears twice, and Reference 71 appears three times.
We thank the reviewer for pointing out the repetition of references in Table 2. To avoid confusion and ensure clarity, we have revised the table so that each citation is used as an example of one method only. We have removed the citations from the other methods, thereby ensuring each reference uniquely highlights a specific technique.
- For Reference 67, it is stated that two geospatial techniques are used: “GIS methodology of spatial join” and “Forward and backward stepwise model”. However, the cited publication states, “The best regression models were selected using the forward and backward stepwise model selection method using the step function in the R-studio software.” Therefore, the “Forward and backward stepwise model” is more likely a statistical method instead of a geospatial technique.
We thank the reviewer for the valuable comment. We acknowledge that the "Forward and backward stepwise model" is a statistical method rather than a geospatial technique, as described in the cited publication. We have revised the manuscript to accurately reflect this distinction. Specifically, we have removed the citation from the "Forward and backward stepwise model" example and clarified the use of the spatial join technique. The spatial join was used to integrate and analyze various datasets, enhancing the geospatial analysis within the study. We included this clarification in the revised version of the document.
- For Reference 71, three geospatial techniques are listed: “Local Moran's I and Getis-Ord Gi*”, “Spatial sampling strategy, Local spatial autocorrelation”, and “spatial sampling strategy across neighborhood-scale sewershed catchments”. Clarification is needed on whether points 2 and 3 are the same. Additionally, in the cited publication, “Local Moran's I and Getis-Ord Gi*” seem to fall within the Local spatial autocorrelation methodology. The authors should clarify these issues.
We thank the reviewer for highlighting the need for clarification regarding the geospatial techniques described in Reference 71. In response to a previous comment, we have selected this study specifically as an example of the use of Local Moran's I and Getis-Ord Gi* and have removed it from other examples. To further clarify, we have provided a more detailed explanation of how these methods were used in the study. Local Moran's I and Getis-Ord Gi* are indeed part of the local spatial autocorrelation methodology, used to analyze the spatial distribution of SARS-CoV-2 viral RNA concentrations in wastewater. We have revised the manuscript to reflect this clarification, ensuring a clear and accurate representation of the methodologies employed in the study.
- In the revised manuscript, Lines 112-114 contain the newly added content, “…… for optimal indoor air quality……”, which seems irrelevant to the other content on WBE in this paragraph.
We have followed the reviewer’s recommendation and removed this sentence in the revised version of the document.
- The reference citations have been improved but still show some inconsistencies in format. For example, Lines 55, 74, 114, 132 and more (citations appear before punctuation in some places and after punctuation in others). These small detail errors should be resolved before acceptance.
We thank the reviewer for noticing this. We have followed the journal’s style and placed all citations before punctuation.